# Adult Postabdomen, Immature Stages and Biology of *Euryommatus mariae* Roger, 1856 (Coleoptera: Curculionidae: Conoderinae), a Legendary Weevil in Europe

**DOI:** 10.3390/insects12020151

**Published:** 2021-02-11

**Authors:** Rafał Gosik, Marek Wanat, Marek Bidas

**Affiliations:** 1Department of Zoology and Nature Protection, Institute of Biological Sciences, Maria Curie–Skłodowska University, Akademicka 19, 20-033 Lublin, Poland; 2Museum of Natural History, University of Wrocław, Sienkiewicza 21, 50–335 Wrocław, Poland; marek.wanat@uwr.edu.pl; 3ul. Prosta 290 D/2, 25-385 Kielce, Poland; zuk55@o2.pl

**Keywords:** beetles, weevils, Curculionoidea, Conoderitae, Coryssomerini, genital structures, larva, pupa, saproxylic species, spruce

## Abstract

**Simple Summary:**

*Euryommatus mariae* is a legendary weevil species in Europe, first described in the 19th century and not collected through the 20th century. Though rediscovered in the 21st century at few localities in Poland, Austria, and Germany, it remains one of the rarest of European weevils, and its biology is unknown. We present the first descriptions of the larva and pupa of *E. mariae*, and confirm its saproxylic lifestyle. The differences and similarities between immatures of *E*. *mariae* and the genera *Coryssomerus*, *Cylindrocopturus* and *Eulechriopus* are discussed, and a list of larval characters common to all Conoderitae is given. The characters of adult postabdomen are described and illustrated for the first time for diagnostic purposes. Our study confirmed the unusual structure of the male endophallus, equipped with an extremely long ejaculatory duct enclosed in a peculiar fibrous conduit, not seen in other weevils. We hypothesize that the extraordinarily long and spiral spermathecal duct is the female’s evolutionary response to the male’s extremely long intromittent organ.

**Abstract:**

The larva and pupa of the saproxylic *Euryommatus mariae* Roger, 1857, the weevil species extremely rare in Europe, are described from Poland. It was reared from galleries in dead branches of a fallen spruce *Picea abies*. The larval morphology is compared with available larval descriptions of other genera of the supertribe Conoderitae, namely, the Palaearctic *Coryssomerus*, and the Nearctic *Cylindrocopturus* and *Eulechriopus*. The specific characters of the male and female postabdomen are described and illustrated, expressing the peculiar structure of endophallus and spermathecal duct, not seen in any other weevil species. A hypothesis regarding the mechanics of mating in this species is proposed. *Euryommatus mariae* is recorded for the first time to occur in China.

## 1. Introduction

The subfamily Conoderinae, Schoenherr, 1833, formerly long known under the commonly accepted junior name Zygopinae Lacordaire, 1866, has been quite recently applied, as the oldest available name, to a much larger group of weevils, including the former subfamilies Baridinae, Ceutorhynchinae, and Orobitidinae, and nowadays comprises over 7500 described species in 940 genera. The former four subfamilies were reduced in rank to supertribes, among which Conoderitae is the second largest, with approximately 2200 species in 212 genera worldwide [1]. However, Conoderitae seems to be the least known taxonomically among the four conoderine supertribes, and the real number of extant species must be several times higher. They are abundant and diverse in samples from rain forest canopies. For example, at least 520 species were counted during an inventory at La Selva Biological Station in Costa Rica by H. Hespenheide [2], the number updated to 559 species until 2009 (H. Hespenheide, personal communication).

While common and hyperdiverse in tropical and subtropical zones, the group is highly under-represented in the Western Palaearctic, in fact by just a single tribe—Coryssomerini—with two genera and four species [3,4]. The strictly Western Palaearctic genus *Coryssomerus* consists of *C*. *capucinus* (Beck, 1817), widespread in Europe and the Caucasus, along with two much less well-known North African species. In the second genus, *Euryommatus* Roger, only its type species *E*. *mariae* Roger, 1857 was originally described from Europe, while eight further species currently attributed to *Euryommatus* live in Eastern Asia, Asia Minor or the Arabian Peninsula [3,4]. The only European species *E. mariae* ultimately turned out to be a Euro-Siberian element, relictual in Europe, but widely distributed throughout Siberia to the Russian Far East, eastern China (the first record here), the Korean Peninsula, and Japan [3,4,5,6,7].

In Europe, *Euryommatus mariae* has always been a mysterious and indeed legendary weevil. Described by Roger [8] from a locality at Rudy near Kuźnia Raciborska (Upper Silesia—the type locality, now in Poland), it was not found again in Silesia or anywhere else in the whole of Poland for over a century. Subsequent records from Europe have been very scarce: All come from the 19th century and only from Austria (Salzburg vic.) [9], and Latvia [10]. The occurrence of *E*. *mariae* in Poland was finally confirmed after 140 years by Stachowiak [11], who collected a single specimen in the Białowieża Primeval Forest (NE Poland). In the 21st century, the species has been found at several new localities in central Europe. In Poland, further specimens were collected in the Świętokrzyskie Mts in central Poland, near the village of Cisów [12]. Recently, the weevil was recorded in southern Bavaria in Germany [13]. At the same time, it was rediscovered in Austria, in the East Tyrol exclave [14]. *Euryommatus mariae* was also recorded in Slovenia, European part of Russia, and Cyprus in recent Palaearctic catalogs [3,4], but these country records are not supported with literature data. Moreover, the latter record is apparently a misidentification, since just an unidentified species of *Euryommatus* is listed from Cyprus in the recent catalog of Curculionoidea of this island [15]. The occurrence of *E. mariae* on this Mediterranean island seems rather unlikely.

Unlike *Coryssomerus capucinus*, which develops on several herbaceous species of the Asteraceae, little is known about the host associations of *Euryommatus mariae* and its congeners. In Austria, it was beaten from ‘Pinus alba’ by Sartorius [9], and probably on this basis, Gerhardt [16,17] presumed its association with fir *Abies alba* Mill. In Siberia, it develops in dying branches of *Abies sibirica* Ledb. [18], but the adults were also collected from *Pinus sylvestris* L. (Legalov, personal communication). In Germany, the weevil was also collected from pine branches, albeit of *Pinus mugo* subsp. *uncinata* (DC.) Domin. In turn, all 28 Polish specimens of *E. mariae*, collected in the forest near Cisów, were obtained using spiral screen-trunk traps (‘Geolas’ trap) deployed on the trunks of spruce *Picea abies* (L.) H. Karst. This may indicate an oligophagous type of host association within the family Pinaceae.

Being familiar with the former sampling area in Cisów, we undertook an investigation there to find immature stages of *Euryommatus mariae* in branches from a dead and fallen spruce *Picea abies*. Laboratory examination of these branches in March 2012 revealed a pupa, found by M. Bidas, who reared it to the adult. The site was revisited in the same month to obtain further material. Here, we give the first morphological description of the larva and pupa of *E*. *mariae* obtained from the above-mentioned spruce branches. The larval characters of *Euryommatus* are compared with available descriptions of some Nearctic species of Conoderitae, including the conifer-associated *Cylindrocopturus furnissi* Buchanan.

## 2. Materials and Methods

The immature specimens used in this study: A premature (probably fourth) instar larva (one ex.), a mature larva (2 exx.), and pupae (♀)(2 exx). All were collected from one site: Cisów, forest district Daleszyce, forest comp. 144, 50.74N/20.86E, 264 m alt., 22.03.2012, branches cut from a fallen spruce trunk, leg. M. Wanat and M. Bidas.

The larvae and pupae are deposited in the collections of the Department of Zoology and Nature Protection, Maria Curie-Skłodowska University (Lublin, Poland).

The ten adult specimens examined are all from the collection of M. Wanat, stored at the Museum of Natural History, University of Wrocław: Poland: Świętokrzyskie Mts: Cisów, o. 144, UTM: DB82, 7–BMw G.Sw. 2, 29–31 VIII 2006, screen traps on spruce, 2 ♂ 3 ♀, leg. J. Borowski; same locality, reared ex dead spruce branches, imagines emerged 6 II 2008 (1 ex.), 17 I 2012 (1 ex.), 1 III 2012, 1 ♂, leg. M. Bidas. China (East): Harbin, 1 ♀. Russia: W Siberia: Altai, Teletzkoe Lake, 12 VII 1975, 1 ♂ 1 ♀, leg. F. I. Opanassenko.

### 2.1. Habitat

The locality near Cisów is a part of large, forested areas within boundaries of the Cisowsko-Orłowiński Landscape Park, undergoing sustainable forest management. It is a natural mixed humid coniferous forest, with dominant pine and spruce, and a small admixture of deciduous trees, mainly birch, alder, and oak (Figure 1A). The forest floor is mossy, and the undergrowth consists primarily of spruce saplings. The site is situated very close to the open peat bog area of the Białe Ługi Reserve. The fallen spruce sampled for *Euryommatus* immatures was a dead tree of medium age, with its side branches still largely covered with loose bark (Figure 1B); hence, it must have lain there for months rather than years, being nearly or quite dead when it fell. Several branches about 3–4 cm in diameter were sawn into pieces and taken to the laboratory in Wrocław. The branches were examined by carefully removing fragments of bark, under which a number of short and simple galleries could be seen. Four galleries contained single larvae, and in one, a pupa of *Euryommatus* was detected in a terminal elongate-oval and only slightly deepened pupal chamber (Figure 1C). The largest larva was left alive in its gallery, where it pupated a few days later. All the immature stages were preserved for morphological studies.

### 2.2. Methods

#### 2.2.1. Morphological Studies—Immature Specimens

All the specimens described were fixed in 95% ethanol and examined under an optical stereomicroscope (Olympus SZ 60 and SZ11) with calibrated oculars. The graduation was scaled up to 1/10 mm. Measurements were made under a 30× magnification. The following measurements of larval instars were made: Body length (BL), body width (BW) (at the third abdominal segment), and width of the head capsule (HW). The pupal measurements included body length (BL), body width (BW) (at the level of the mid legs), head width (HW) (at the level of the eyes), length of rostrum (RL), and width of pronotum (PW). The drawings and outlines were made using a drawing tube (MNR–1) installed on a stereomicroscope (Amplival Pol-d, Carl Zeiss Jena, Germany). and processed by computer software (Corel Photo-Paint X7, Corel Draw X7 (Corel Inc. Austin, TX, USA).

Slide preparation basically follows May [19]. The larvae selected for study under the microscope were cleared in 10% potassium hydroxide (KOH), then rinsed in distilled water and dissected. After clearing, the head, mouthparts, and body (thoracic and abdominal segments) were separated and mounted on permanent microscope slides in Faure–Berlese fluid (50 g gum arabic and 45 g chloral hydrate dissolved in 80 g of dissolved water and 60 cm^3^ of glycerol) [20].

The photographs were taken using an Olympus BX63 microscope and processed with Olympus cellSens Dimension software. The larvae selected for SEM imaging (scanning electron microscope) were first dried in absolute ethanol (99.8%), then rinsed in acetone, treated by CPD (Critical Point Drying), and finally gold-plated. TESCAN Vega 3 SEM was used to examine selected structures. The general terminology and chaetotaxy follow Anderson [21], May [19], Marvaldi [22,23,24,25], and Skuhrovec et al. [26]; the antennae terminology follows Zacharuk [27].

#### 2.2.2. Morphological Abbreviations

**Abd. I–X—**abdominal segments 1–10, **Th. I–III**—thoracic segments 1–3, **at**—antenna, **clss**—clypeal sensorium, **ds**—digitiform sensillum, **st**—stemmata, **Se**—sensorium, **sa**—sensillum ampullaceum, **sb**—sensillum basiconicum, **snp**—sensillae pore, **ss**—sensillum styloconicae, **tra**—terminal receptive area, **lr**—labral rods, **ur**—urogomphus; setae: ***als***—anterolateral, ***ams***—anteromedial, ***as***—alar (larva), ***as***—apical (pupa), ***cls***—clypeal, ***d***—dorsal (pupal abdomen), ***des***—dorsal (larval head), ***dms***—dorsal malar, ***ds***—discal (pupal prothorax), ***ds***—dorsal (larval abdomen), ***eps***—epipleural, ***es***—epistomal, ***eus***—eusternal, ***fs***—frontal, ***les***—lateral epicranial, ***ligs***—ligular, ***lrs***—labral, ***ls***—lateral, ***lsts***—laterosternal, ***mbs***—malar basiventral, ***mds***—mandibular, ***mes***—median, ***mps***—maxillary palp, ***pda***—pedal, ***pds***—postdorsal, ***pls***—posterolateral, ***pes***—postepicranial, ***pfs***—palpiferal, ***pms***—postlabial, ***prms***—prelabial, ***prns***—pronotal, ***prs***—prodorsal, ***ps***—pleural, ***sls***—super lateral, ***sos***—superorbital, ***ss***—spiracular, ***stps***—stipal, ***sts***—sternal, ***ves***—ventral, ***vms***—ventral malar, ***vs***—vertical. **HW**—head width, **BL**—body length, **BW**—body width, **RL**—rostrum length, **PW**—pronotum width.

#### 2.2.3. Morphological Studies—Adult Specimens

Images of adult specimens were taken with a Leica M205C stereomicroscope and attached digital camera JVC KYF75. The images obtained were combined using the AutoMontage software of Syncroscopy (Cambridge, UK), and enhanced using Adobe Photoshop CS2 program. The aedeagus was photographed under transmitted light using the same equipment, while the details of the conduit of the ejaculatory duct were illustrated using a Nikon Eclipse *Ni* compound microscope with attached Nikon D7500 camera, and the stacks were combined using the Helicon Focus software (ver. 7.5.1 Pro, Helicon Soft Ltd., Kharkiv, Ukraine).

Genitalia preparations were made according to the standard method after a maceration of the separated abdomen for 5–10 min in hot KOH solution. Membranous structures were stained in a glycerol solution of chlorazol black. After rinsing in distilled water, further separation and examination of the terminal segments and genitalia were carried out in pure glycerol on a microscope slide, under both stereoscopic and compound microscopes. Only the spermatheca was first photographed in distilled water, to illustrate its natural shape, before the eventual collapse of its wall after transfer to glycerol. After the study, all the parts were stored in glycerol in a microvial pinned beneath the specimen, while the abdominal ventrites were glued onto a card adjacent to the specimen.

## 3. Results

### 3.1. Description of the Larva of Euryommatus Mariae

Premature larva (4th instar): HW: 0.70 mm (rest of the body deformed).

Mature larva (5th instar): HW: 0.95 mm; 0.95 mm; BL: 3.35 mm; 3.50 mm; BW: 1.60 mm; 1.40 mm.

*General habitus* (Figure 2). Living larva is pure white, with a pale brown head capsule. Its body is rather stout, curved, and rounded in the cross section. The prothorax small, with a slightly pigmented pronotal shield; meso- and metathorax almost equal in size. The meso- and metathorax are each divided dorsally into two lobes (prodorsal lobes very small, postdorsal lobes prominent). The pedal lobes of thoracic segments are isolated and conical. Abdominal segments I and II are of similar, medium size, and segments III–V are the biggest. The next segments taper towards posterior body end. Abdominal segments I–VII are narrow, and the prodorsal lobe, and always well-developed postdorsal lobe is divided into three parts (first and third wide, second narrow). Segment VIII, with narrow prodorsal lobe and wide postdorsal lobe, is divided into two folds equal in length. The epipleural lobes of segments I–VI are slightly conical, and prominent on segments VII and VIII. The laterosternal and eusternal lobes of segments I–VIII are conical and weakly isolated. Abdominal segment IX is undivided dorsally. Abdominal segment X is divided into four lobes of almost equal size. The anus is situated terminally. Only the pronotal sclerites are brownish, and the remaining thoracic and all abdominal segments are white or greyish. The body cuticle is densely covered with asperities, taking a form of thorns (Figure 3A–D), and on dorsal parts of abdominal segments, additionally, with peculiar elongate plate-like asperities—each parallel to long body axis and bicornuate, arranged in transverse rows (Figure 3C). All spiracles are bicameral; thoracic (Figure 4A) placed latero-ventrally are on the prothorax; abdominal spiracles (Figure 4B) are placed medio-laterally on segments I–VIII.

*Chaetotaxy* (the number of setae is given for one side of the body) (Figure 5A–C). Setae of various lengths, from very long to minute, always in the form of hairs. Thorax (Figure 5A): Prothorax with eight long and two medium *prns* (eight on pronotal sclerite, next two above the spiracle), two long *ps*, and two medium *eus*. Meso- and metathorax each with one medium *prs* and four *pds* of various length (first long, second medium, third, and fourth long), alar area with one medium *as*, three *ss* of various length (two long and one min), one long *eps*, one long *ps*, and one medium *eus*. Pedal areas of thoracic segments each with six variably sized *pda*. Abdomen (Figure 5B,C): Segments I–VII with one short *prs*, five *pds* (first, third, and fifth medium, second, and fourth short to minute), two *ss* (first minute, second medium), two *eps* (one medium and one min), two *ps* (one medium and one min), one medium *lsts* and two medium *eus*. Abdominal segment VIII with one short *prs*, three very long *pds*, and *one* long *ss* (chaetotaxy of next folds as on segment VII). Abdominal segment IX with three very long *ds*, three *ps* (one long, one medium, and one min), and two medium *sts*. Abdominal segment X with three *ts* (two short and one minute) on each lobe.

*Head and antenna* (Figure 6A,B and Figure 7). Head pale testaceous (Figure 6A), slightly narrowed bilaterally; endocarina present, endocarinal line short, one-third as long as frons; frontal sutures distinct along entire length up to antennae; single pair of stemmata (st) in the form of prominent dark pigmented spots with a convex cornea, placed anterolaterally (Figure 6B). Hypopharyngeal bracon without median sclerome. Setae of head of various length, very long to minute. Cranial setae: *des_1_* elongated, placed medially, *des_2_* elongated, placed posterolaterally, *des_3_* elongated, placed above frontal suture, *des_4_* slightly shorter than other *des*, placed anteromedially, *des_5_* elongated placed anterolaterally, *fs_1_* as long as *des_4_*, placed posteriorly, *fs_2_* short, placed medially, *fs_3_* elongated, placed anteromedially, *fs_4_* and *fs_5_* elongated, placed anterolaterally, close to epistome, *les_1_* and *les_2_* as long as *des_4_*, two *ves* short, postepicranial area with four min *pes*. Antennae (Figure 7) with oblique position on each side at anterior margin of head; membranous basal segment convex, semi-spherical, bearing conical, moderately elongated sensorium and seven sensilla: Three basiconica (sb), three styloconica (ss), and one ampullaceum (sa).

*Mouth parts* (Figure 8A,B, Figure 9, Figure 10A–D, and Figure 11A,B). Clypeus (Figure 8A,B) approximately 3 × wider than long, *cls_1_*_–*2*_ elongated, with sensillum (clss) between them, all placed posterolaterally. Anterior margin of clypeus straight. Labrum (Figure 8A,B) approximately 2× wider than long, anterior margin almost semicircular; *lrs_1_* medium, placed anteromedially, *lrs_2_* elongated, placed medially and *lrs_3_* elongated, placed posterolaterally. Epipharynx (Figure 8A,B) with three digitate *als*, of various length, three finger-like *ams*: *ams_1_* elongated, *ams_2_* thin, *ams_3_* short and curved; *mes_1–2_* equal in size, robust. Labral rods (lr) elongated, more sclerotized at apex and wider towards base, only slightly converging posteriorly. Sensillae pores (snp) arranged in the middle of epipharynx: Two pairs close to *mes*, next two pairs in the middle of labral rods. Surface of epipharynx between labral rods covered with thorn-like asperities. Mandibles (Figure 9) with two apical teeth of unequal height, the inner one subapical and much smaller. Cutting edge between apex and middle of mandible with additional protuberance. Setae: *mds_1_* min, *mds_2_* medium, both placed laterally in shallow pits. Maxillolabial complex (Figure 10A–D) on stipes with one elongated *stps*, two elongated *pfs*, and one min *mbs* plus sensillum. Mala with row of seven digitate, almost equally-sized *dms*, and five *vms* (two medium and three short). Maxillary palpi with two palpomeres; basal palpomere slightly wider than distal one. Length ratio of basal and distal palpomeres almost 1:1. Basal palpomere with one short *mps* and two pores, distal palpomere (Figure 11A,B) with one pore, one digitiform sensillum (ds), and a group of 11 apical sensillae (ampullaceae) on terminal receptive area (tra). Dorsal parts of mala partially covered with fine asperities. Labium with prementum cup-shaped, with one medium *prms* placed medially. Ligula concave, semicircular at margin, with two min *ligs.* Premental sclerite trident-shaped (median branch weakly sclerotized), posterior extension with elongated, sharp apex; postmentum rather narrow, membranous, triangular, with three elongated *pms*: *pms_1_* situated posterolaterally, *pms_2_* mediolaterally and *pms_3_* anterolaterally. Labial palpi two-segmented; basal palpomere wider and longer than distal one. Length ratio of basal and distal palpomeres almost 1:0.7. Each palpomere with single pore, distal palpomere with a group of nine apical sensillae (ampullaceae) on terminal receptive area. Lateral and posterolateral parts of labium covered with prominent asperities.

### 3.2. Description of the Pupa of Euryommatus Mariae

Female: BL: 4.00; 4.00; BW: 2.10; 2.50; HW: 0.75; 0.85; RL: 0.80; 0.80; PW: 1.15; 1.25 (all in mm).

*General habitus and chaetotaxy* (Figure 12A–F and Figure 13A–C). Body white, slender, cuticle covered with fine asperities, smooth only on head and pronotum (Figure 12A–F). Rostrum elongated, 2.8× as long as wide, reaching mesocoxae. Pronotum 1.5× wider than long, rounded laterally. Mesonotum wider than metanotum. Abdominal segments I–III of equal length, segments IV–VI tapering gradually towards the end of the body, segment VII semicircular, segment VIII narrow, segment IX terminal, with urogomphi (ur) medially situated, slightly recurved, elongated, covered with asperities, each with sclerotized, anchor–like apex. Spiracles placed dorso-laterally on abdominal segments I–VI, functional on segments I–V, vestigial on segment VI. Chaetotaxy (setal numbers are given for one side of the body): Setae variable in size, hair-like on head and rostrum, thorn-like on thorax and abdomen, all placed on prominent protuberances. Head with two *pas* of various sizes and three *os* of unequal length, rostrum with one *rs* (Figure 13A–C). Pronotum with one *as*, one *ls*, one *sls*, one *ds*, and three *pls* almost equal in size. Meso- and metathorax with two short setae placed medially on dorsum. Abdominal segments I–VI with three short setae (*d_1_* placed medially, *d_2_*, and *d_3_* more laterally). Segment VII with three (segment VIII with 2) thorn-like, robust setae, placed on elongated protuberances. Each side of segment IX with single robust, thorn-like seta placed dorsolaterally, and urogomphus placed dorsomedially. Lateral and ventral parts of abdominal segments I–VIII without setae. Each femur with two elongated, hair-like setae (Figure 13A–C).

### 3.3. Description of the Adults of Euryommatus Mariae

In Europe, *Euryommatus mariae* is the only member of its genus and a morphologically distinct species, included in several identification keys [28,29,30]. This small weevil (body length 3.3–4.2 mm in studied specimens) is easily and immediately distinguishable from all other European weevils by its habitus (Figure 14A): The abdominal ventrites rising steeply posterad and covered with flattened elytra, weakly sloping apicad (Figure 14B); the hypognathous head with enlarged eyes covering its entire dorsum and separated just by a thin integumental septum no thicker than twice the diameter of a minute ommatidium; the prosternum lacking any trace of a rostral canal; the large and prominent mesepimera; the broad metanepisterna; the round scutellar shield surrounded by a deeply impressed ring; the posterior margins of abdominal ventrites 2–4 expanded vertically, bare and shiny (Figure 14C); the large, sharp tooth on the fore femur and the vestigial teeth on the mid- and hind femora, and all the tibiae shorter than the femora.

The male and female terminalia of *E*. *mariae* have never been described or illustrated, albeit Wanat [31] did mention the unusually long ejaculatory duct, supplying a photo of aedeagus in support of his statement. Examination of the postabdomen revealed several characters which may serve to distinguish this species from its congeners. One of them, the structure of the endophallus and especially of the ejaculatory duct, is unusual and probably unknown in other weevils. The structure of the postabdomen is described and illustrated below for diagnostic purposes.

Male abdominal ventrite 5 apically with a paired tuft of protruding setae. Tergite VII strongly transverse, ca 2.2–2.3 × as broad as long in the middle, with short anterior arms, apically shallowly emarginate, in its entirety heavily sclerotized and coarsely punctate, with large lateral wing-folding patches (Figure 14E). Pygidium (tergite VIII) exposed from elytra in repose, apically with a large impression bordered by sharp carinae, an impression in the middle with paired, small, lighter windows, on sides with dense raised scales (Figure 14F,G). Hemisternites VIII, large and broadly connate medially (Figure 14D). Spiculum relictum on membrane fold between sternites VIII and IX absent, even as a membranous process. Tegmen with apodeme much shorter than forked basal piece; parameral lobes long and narrow, ca 2/3 as long as pedon, each with median sclerotized stripe. Pedon in profile sharply hooked apically (Figure 14H); ostium with large paired sclerites (Figure 14I). Membranous endophallus small and fully contained in pedon. The whole ejaculatory duct in a peculiar, sclerotized conduit-like sheath composed of dense fibrae and unstainable with Chlorazol black (Figure 14H), a few times longer than the beetle’s body itself (!), in repose looped many times outside the penis proper, tightly folded into a package adjoining ventral side of pedon and its apodemes; the rigid sheath enters the pedon through the basal foramen and near the orifice joins an elongate brush of large and partly radiate spines.

Female. Abdominal ventrite 5 only with appressed scales. Tergite VII trapeziform, only 1.5–1.6 × as broad as long, broadly rounded apically, in its entirety heavily sclerotized and coarsely punctate, with wing-folding patches smaller than in male, with a complete transverse carina separating apical part with much smaller and sparser punctures bearing longer setae (Figure 15A). Tergite VIII sub-pentagonal in shape, markedly tapering from basal third to apex, wholly concealed beneath tergite VII in repose, weakly and evenly sclerotized except for a narrow, clear median line in proximal two-thirds; apical one-third densely setose, with setae progressively longer towards tergum apex (Figure 15B). Spiculum ventrale (Figure 15C) with short apodeme; sternal plate large, with a median sclerotized fork and largely desclerotized margins, bearing about a dozen long setae on apices of fork arms. Ovipositor consisting of subrectangular coxites about twice as long as broad and tightly joined to the folded membrane (Figure 15D) and styli attached latero-apically to coxites, large, strongly elongate, not less than 0.4 × as long as coxite, numerous long setae along entire apex, i.e., 0.4 × length of stylus. Vagina composed of simple, but thick and multi-folded membrane. Bursa narrow, unilobed, composed of simple fine membrane. Spermatheca small, C-shaped, with spherical corpus lacking prominences and narrower cornu (Figure 15E); gland relatively large, elongate; spermathecal duct very long, unsclerotized, in its entirety tightly spiral (Figure 15D,E).

## 4. Discussion

The host associations and basic details of larval development have been identified in just one European species from each genus within the tribe Coryssomerini. *Coryssomerus capucinus* lives on several herbaceous plants from the genera *Tripleurospermum* Sch. Bip., *Achillea* L., *Chrysanthemum* L., and *Anthemis* L. (Asteraceae); its larva develops in the root neck and pupates in the soil [32,33]. The biology of *Euryommatus mariae* is quite different, however: It has saproxylic, subcortical larvae associated with gymnosperm trees. Such an association between conifers and saproxylic larvae is unknown in any other Palaearctic member of the supertribe Conoderitae. However, analogous host associations with coniferous trees and wood-boring larvae have been well documented in several Nearctic species of the genus *Cylindrocopturus*, classified in the tribe Zygopini [34], and distributed along Pacific coast of North America [35]. According to Furniss and Carolin [36], they attack the twigs and boles of various conifers, including pine (*Pinus* L.), true fir (*Abies* L.), Douglas fir (*Pseudotsuga* Carrière), larch (*Larix* Mill.), and hemlock (*Tsuga* Carrière). The Douglas fir twig weevil *Cylindrocopturus furnissi* Buchanan develops on *Pseudotsuga menziesii* (Mirbel) Franco [37], while *C*. *eatoni* Buchanan, known as the pine reproduction weevil, attacks primarily ponderosa and Jeffrey pines (*Pinus ponderosa* Douglas ex C.Lawson, *P*. *jeffreyi* Balf.). Although associated with conifers in much the same way as *E*. *mariae*, the biology of these Nearctic *Cylindrocopturus* species is significantly different. They both attack the branches and boles of saplings and young trees with green foliage and develop in living wood, their larvae often killing them by destroying the phloem and cambium, thus cutting off sap transport. The larvae of *C*. *eatoni* can be found in all woody parts of pine saplings, even in the rootstock some centimetres below ground level, but the top of the stem and the upper branches are infested much more abundantly [38]. *C. furnissi* also attacks small living trees of Douglas fir, and in older ones it evidently prefers the previous four years’ growth for oviposition, frequently killing scattered small branches [36]. In both *Cylindrocopturus* species, the eggs are laid individually in holes chewed by the female in living bark—globules of resin produced by the tree reveal their locations. Their larvae bore galleries in cortical tissue, phloem or even pith, and often pupate there in thin twigs, while in thicker branches and the trunk, mature larvae usually work their way into the outer layer of wood for pupation. The different habits of *Euryommatus mariae* can be inferred from the data collected at the Cisów site. There, the weevil seems to be confined to dead or dying spruce branches, presumably weakened and dying lower laterals, subsequently appearing on older trees that retain their bark intact for quite a long time after death. Unlike the American conifer beetles *Cylindrocopturus*, *E*. *mariae* is a saproxylic species utilizing the outermost layer of dead or dying wood.

Apart from this basic difference in diet, many other aspects of the biology of *E*. *mariae* appear to resemble that of the above-mentioned *Cylindrocopturus* species. Their development takes a year with a winter break [36,37,38], an aspect that also appears to hold true for *E*. *mariae*, where the larva is the diapausing stage. In its phenology, too, *E*. *mariae* seems to follow *C*. *furnissi* in particular. This latter species starts to make feeding holes in the bark of small Douglas fir branches, mating, and laying eggs after mid-June, with adults leaving the pupal chambers by the beginning of August [37]. The emergence of the adults of *C*. *eatoni* begins about a month earlier, already in late May, and peaks in mid-June, when the weevils can be found feeding abundantly on pine needles [38]. Although the larvae collected in Cisów on 22 March pupated in the same month, and one adult emerged on 1 March from material preserved early in winter, they were all reared under laboratory conditions at room temperature, which could have speeded up metamorphosis considerably. The numerous adults of *E*. *mariae* obtained by Rutkiewicz [12] from the screen-trunk traps deployed in Cisów are all dated to the end of August. Moreover, the recent collections of this species took place in mid-August in Germany [13], and in mid-July in Austria [14].

The morphology of immatures of species from the tribe Coryssomerini remains poorly studied. Only the paper by Urban [39] contains some basic, unillustrated information about the larva and pupa of *Coryssomerus capucinus* (Beck, 1817), subsequently published by Scherf [32]. Urban [39] highlighted the larval cuticle densely covered with hook-like asperities, six variously long setae on each of the pedal lobes, the two-segmented labial palpi, and bifid mandibles. Unfortunately, other characters are either very common or were described inaccurately. But it is noteworthy that the larval cuticle of *Euryommatus mariae* exhibits a very similar structure to that of the larva of *C. capucinus*. Urban’s [39] description of the pupa of *C. capucinus* did not contribute any significant information.

Hinz and Müller-Schärer [40] reported five larval instars of *C. capucinus* based on the width of the head capsule. According to these authors, the head width of the premature larva is 0.70 mm, while in the mature larva it is 0.93 mm [40]. Both values resemble the head measurements of *E. mariae* (0.70 mm and 0.95 mm). From this, one may presume that the larval development of *E. mariae* also involves five instars. Despite the highly precise measurements, the paper by Hinz and Müller-Schärer [40] does not contain any other information about the larval morphology. Thus, the descriptions of the larva and pupa of *E. mariae* given here are the first complete, illustrated information on immatures of the tribe Coryssomerini.

It seems that the immatures of most genera belonging to the subfamily Conoderinae Schoenherr have yet to be described. The only exceptions are a few species from the tribe *Zygopini* (regarded as pests of silviculture): The genus *Cylindrocopturus—C. crassus* van Dyke, Keifer [41], *C. quercus* (Say) [42], and *C. furnissi* Buchanan [43], and a single species from the genus *Eulechriopus—E. gossypii* Barber Böving [44]. Analysis of these papers is sometimes difficult, however, because of the different nomenclature used in them and the lack of descriptions of certain structures: For instance, the abdominal setae of *E. gossypii*, according to Böving [44] are extremely small and impossible to count accurately.

Compared with the larvae of other Conoderinae species, the larva of *E. mariae* reveals some important, original features, different from *Cylindrocopturus*, and in some cases, *Eulechriopus*, such as: (1) Single *as* (vs. two *as*); Abd. VIII with three *p**ds* (vs. two *ds*); Abd. IX with three *ds* and three *ps* (vs. two *ds* and two *ps*); (3) head rounded (vs. narrowed bilaterally); (4) abdominal segments dorsally with transverse rows of elongate plate-like asperities, each parallel to long body axis and with anterior and posterior denticle (vs. only the simple, thorn-like asperities); (5) erect setae on thorax and abdominal segments VIII and IX very long, evidently longer than on the remaining abdominal segments (vs. setae on all body segments subequally short).

The best visible common characters of the genera *Cylindrocopturus* and *Euryommatus* include the extended head; the single pair of ocelli placed close to the antenna; the frons with five setae; the mala with seven *dms,* and five *vms*; the clypeus with very long setae; the structure of the spiracles; the dorsal folds of the abdominal segments I–VII divided into three lobes; the well-developed abdominal setae; the prothorax with 10 (11 on *C. furnissi*) *prms*; each pedal lobe with six *pda*; the meso- and metathorax each with one *prs* and four *pds*; each of abdominal segments I–VII with one *prs*, five (four on *C. quercus*) *pds*, two *ss*, two *eps*, and two *ps*.

On the other hand, the larva of *E. gossypii* displays many original characters, different from both *Cylindrocopturus* and *Euryommatus*, above all the strongly retracted head and the structure of the last abdominal segments closely resembling the type “B” described by van Emden [45], found, for example, on *Tanymecus*, *Strophosoma* and *Philopedon* [46]. Moreover, the larva of *Eulechriopus* has the dorsal folds of abdominal segments I–VII divided into two lobes, two pairs of ocelli, and the clypeus without setae [44].

The larval characters common to all Conoderinae genera are (1) endocarina present; (2) conical, moderately elongated antennal sensorium; (3) epipharynx with three *als*, three *ams*, and two *mes*; (4) labral rods very elongated, slightly converging posteriorly; (5) bifid mandible; (6) two-segmented labial palpi; (7) dorsal part of the body densely covered with asperities; (8) all spiracles bicameral.

Based on head width measurements of *C. quercus*, Piper [42] reported three larval instars: The first 0.18–0.19 mm, the second 0.23–0.24 mm, and the third 0.50–0.72. Hence, GF takes a value of 1.35 between the first and second instars (1.32 on *C. capucinus*), but 2.60 between the second and third instars (1.41 on *C. capuccinus*). Moreover, GF measured between the smallest and largest larvae, assessed by Piper [42] as third instars, takes a value of 1.44, which is similar to the GF estimated between the fourth and fifth instars of *C. capucinus* [40]. Hence, the measurements performed with the GF proposed by Dyar [47] indicates (most probably) the existence of five larval instars in *C. quercus*.

According to Böving [44], the pupae of *E. gossypii* and *C. crassus* are visibly similar in shape; this may also apply to the pupae of *C. furnissi* [43], *C. quercus* [42], and in some ways to *E. mariae*, mainly because of the great morphological similarity of the adult stages of those species. However, the lack of accurate drawings and precise nomenclature of the chaetotaxy does not permit any far-reaching inferences to be drawn. Anderson [43] described the urogomphi of *C. furnissi* as being placed laterally, “slightly sclerotized at the tip and terminating in two or three minute projections”. It is hard to say whether the structure described by Anderson [43] is something that corresponds to the urogomphi of *E. mariae.* The pupa of *E. mariae* has quite elongated urogomphi with anchor-like apices, placed dorsomedially, which is rather uncommon in weevil pupae and well visible. Thus, it does not seem possible that such a crucial feature has been overlooked in the descriptions of *Cylindrocopturus* and *Eulechriopus.* Ultimately, however, it is impossible to distinguish any features of taxonomic importance for Conoderitae, based on existing descriptions of these pupae.

The study of the adult postabdomen has yielded several morphologically and evolutionarily noteworthy discoveries. The unusually long ejaculatory duct protected by a fibrous sheath is a unique character of *E*. *mariae*, not found in its congeners available for this study, or in *Coryssomerus* from the same tribe. However, an unidentified *Euryommatus* sp. from Fethiye in southern Turkey (coll. M. Wanat) possesses in its endophallus a flagellum-like pipe of similar fibrous structure, though incomparably shorter, not exceeding the tips of the penile apodemes. On the other hand, the ejaculatory duct in another *Euryommatus* sp. from the Myohyang Mts. in North Korea (coll. Museum and Institute of Zoology, Polish Academy of Sciences, Warsaw) is completely different, broad and tape-like, and unprotected by any additional sheath. The functionalism of such a bizarre ejaculatory duct and the mechanics of copulation in *E*. *mariae* remain unknown and are difficult to imagine. The only observed peculiarity in the structure of the female postabdomen, eventually in response to the male modification, is the extraordinarily long and spiral spermathecal duct. This might suggest the penetration and spreading of this duct by the peculiar rigid male conduit. This hypothesis is supported by the fact that, as shown in (Figure 15D), the spermathecal duct thickens progressively towards the bursa and is visibly broader near its opening than at its spermathecal end. If confirmed by direct observation, possibly based on the immediate freezing of mating individuals, this would be a striking example of evolutionary competition between sexes.

## 5. Conclusions

The larvae and pupae of *Euryommatus mariae* (Coryssomerini) were reared from dead spruce *Picea abies* branches collected in central Poland. Its association with coniferous trees from the family Pinaceae, highlighted in the literature, is thus confirmed, even though spruce has never actually been mentioned as a potential host of this weevil. The species turned out to be saproxylic, in contrast to several Nearctic species of *Cylindrocopturus* (Zygopini) that develop in the living tissues of various American conifers and have a similar life cycle and phenology.

The very limited knowledge of the morphology of the immature stages of Conoderitae hinders comparisons and estimates of differences. The larva of *E. mariae* shares the cuticle densely covered by thorn-like asperities with both *Coryssomerus* and the members of *Cylindrocopturus*, and probably also with *Eulechriopus* (both Zygopini), although in the last-mentioned genus, the description of this character in *E. gossypii* in Böving [44] was inaccurate. This peculiar character, thus, appears to be common to a wider group of genera of Conoderitae. The unique larval characters of *E. mariae*, not recorded in the other genera being compared here, are the single alar seta (*as*) (vs. two *as*), the abdominal segment VIII (Abd. VIII) with three postdorsal setae (*pds*) (vs. two dorsal setae (*ds*)), and the rounded head (vs. narrowed bilaterally), and elongate, bicornuate plate-like asperities arranged in transverse rows on the dorsum of abdominal segments, the character not known in other weevil larvae. The most easily observed diagnostic character of *E. mariae* is the presence of numerous long, erect setae on thoracic segments I–III and abdominal segments VIII–IX. These were not reported in any of the species compared, in which all the dorsal setae are short to poorly discernible. This may be a consequence of saproxylic mode of larval life of *E. mariae*, in the galleries bored under loose bark of dead tree branch, where the larva has to control more space than the larvae of other compared species boring a narrow channel in tight living tissues.

*Euryommatus mariae* is widely distributed in the taiga zone of Eastern Palaearctic [3,4,5,6,7], whilst in Europe it seems to be very local and relict species found in just a few spots of natural coniferous forests in both the mountains (the type locality, Austria, Germany), and the lowlands (Latvia, Poland). Its disjunct distribution summarized in Figure 16, the apparent confinement to large and natural or semi-natural forests of boreal type, and the evidenced association with deadwood, all support well the selection of *E. mariae* for the group of umbrella species in forest conservation [48]. The species is sporadically collected, and despite a very large area of distribution, the number of documented records is very low, either in Europe or Asia. It makes uncertain the gap in its range between Siberia and Europe, resulting from the hitherto literature records.

The unusually long male ejaculatory duct protected with a fibrous sheath is a unique character of *E*. *mariae*, probably unknown in other weevils (Curculionoidea). This contrasts with the extremely long female spermathecal duct, which, moreover, is spiral along its whole length. These two striking characters seem to be correlated and suggest the possible deep penetration of the spermathecal duct by the male with his half-rigid ejaculatory duct sheath; but this will have to be confirmed by direct observation.

## Figures and Tables

**Figure 1 insects-12-00151-f001:**
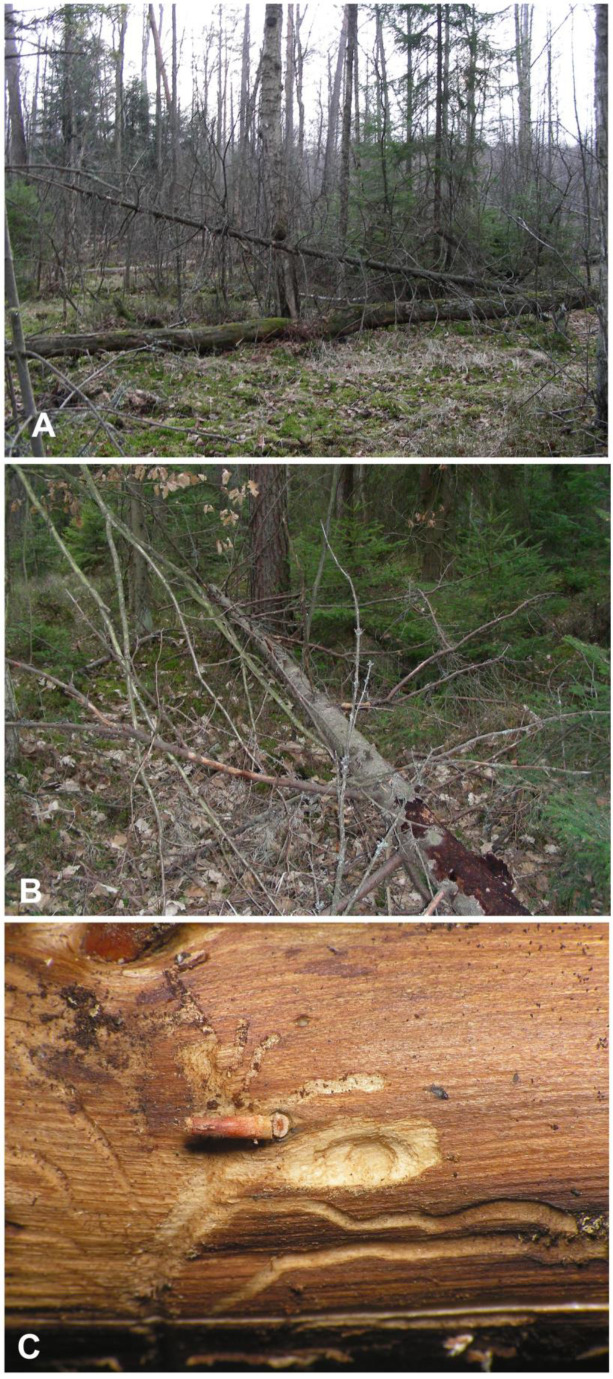
Biotope of *Euryommatus mariae*. (**A**) Forest in Cisów (Świętokrzyskie Mts., Poland); (**B**) dead spruce hosting the larvae; (**C**) larval gallery.

**Figure 2 insects-12-00151-f002:**
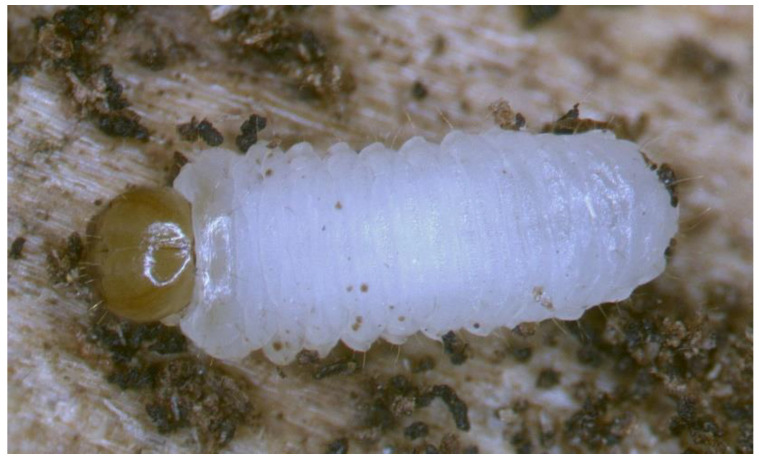
*Euryommatus mariae* live larva photographed in a gallery under a Leica stereoscopic microscope, length 3.8 mm.

**Figure 3 insects-12-00151-f003:**
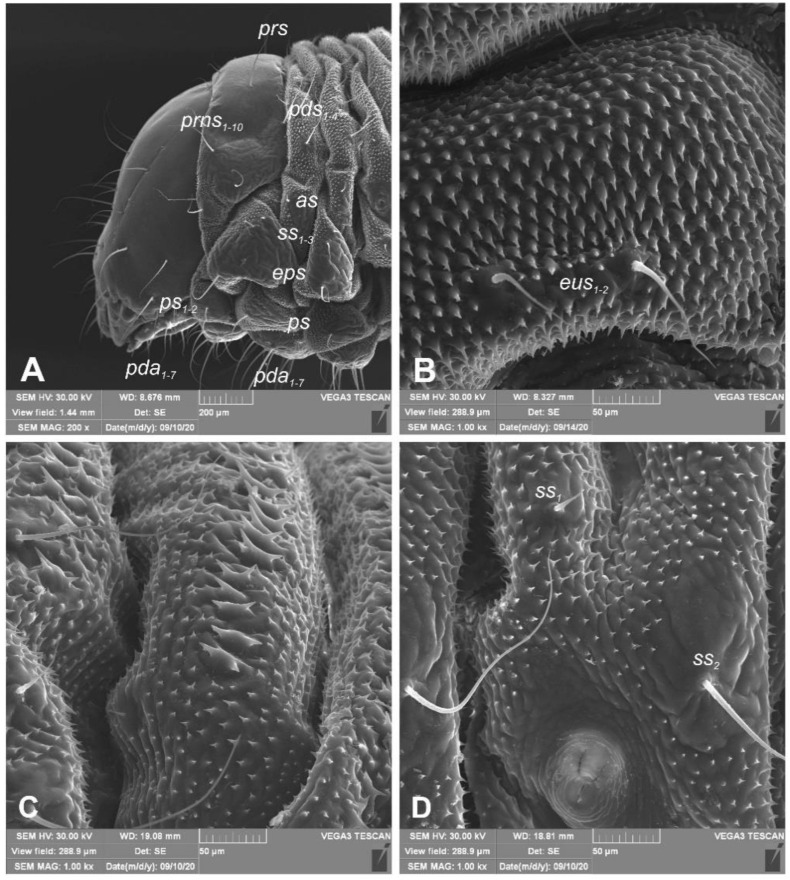
*Euryommatus mariae* mature larva, habitus, and cuticle (SEM micrographs, TESCAN ORSAY HOLDIND, Brno, Czech Republic). (**A**) Lateral view of head and thorax; (**B**) ventral view of the abdominal segment I; (**C**) dorsal view of the abdominal segment I; (**D**) lateral view of the abdominal segment I (setae: *as*—alar, *ps*—pleural, *eps*—epipleural, *eus*—eusternal, *pda*—pedal, *pds*—postdorsal, *prns*—pronotal, *prs*—prodorsal, *ss*—spiracular).

**Figure 4 insects-12-00151-f004:**
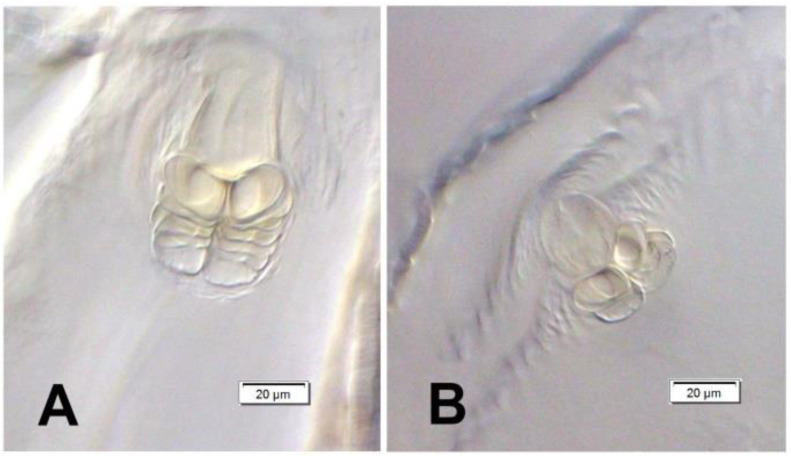
*Euryommatus mariae* mature larva, spiracles. (**A**) Spiracle of prothorax; (**B**) spiracle of abdominal segment I.

**Figure 5 insects-12-00151-f005:**
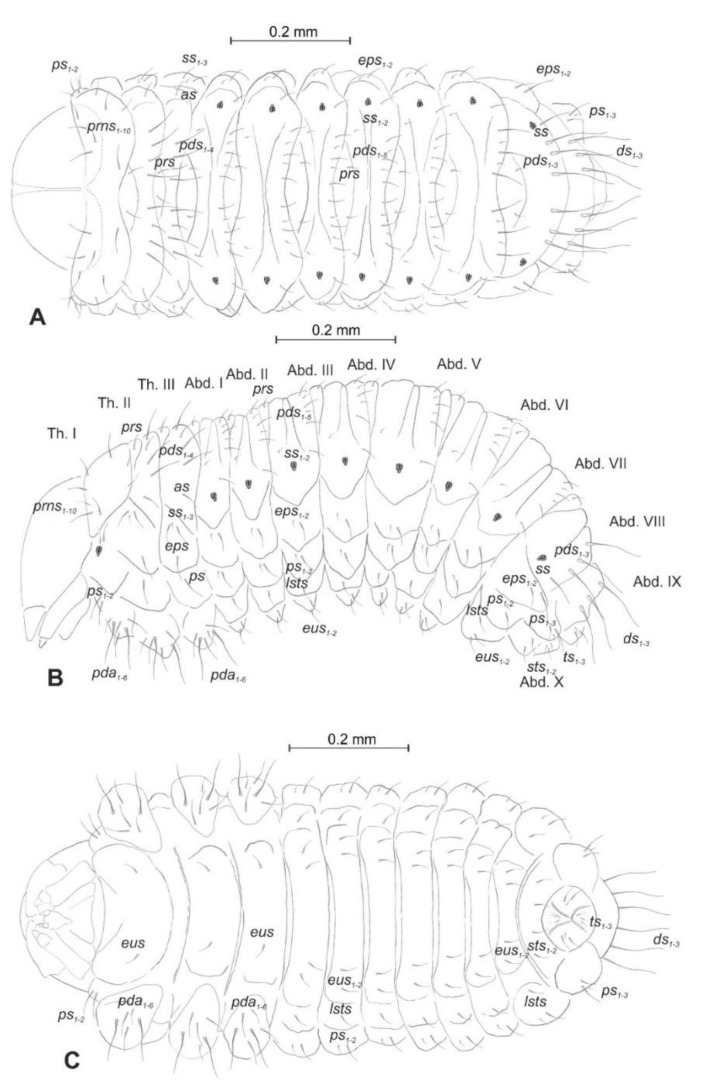
*Euryommatus mariae* mature larva, habitus, and chaetotaxy. (**A**) Dorsal view; (**B**) lateral view; (**C**) ventral view (Th. I–III—thoracic segments 1–3, Abd. I–X—abdominal segments 1–10, setae: *as*—alar, *ps*—pleural, *eps*—epipleural, *ds*—dorsal, *lsts*—laterosternal, *eus*—eusternal, *pda*—pedal, *pds*—postdorsal, *prns*—pronotal, *prs*—prodorsal, *ss*—spiracular, *sts*—sternal, *ts*—terminal).

**Figure 6 insects-12-00151-f006:**
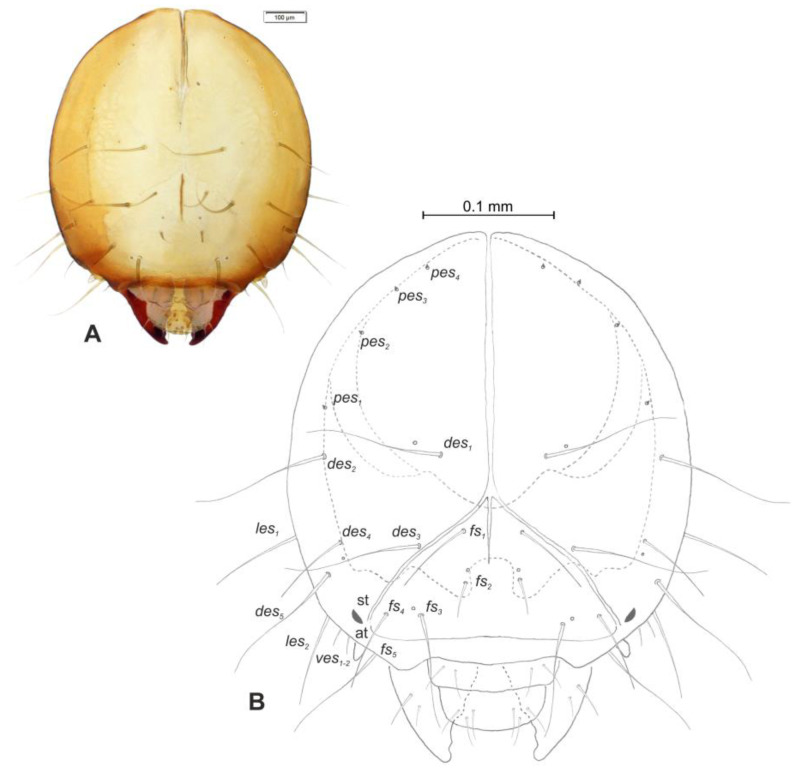
*Euryommatus mariae* mature larva, head, frontal view. (**A**) Frontal view, photo; (**B**) frontal view, scheme (at—antenna, st—stemmata, setae: *des*—dorsal epicranial, *fs*—frontal, *ls*—lateral epicranial, *pes*—postepicranial).

**Figure 7 insects-12-00151-f007:**
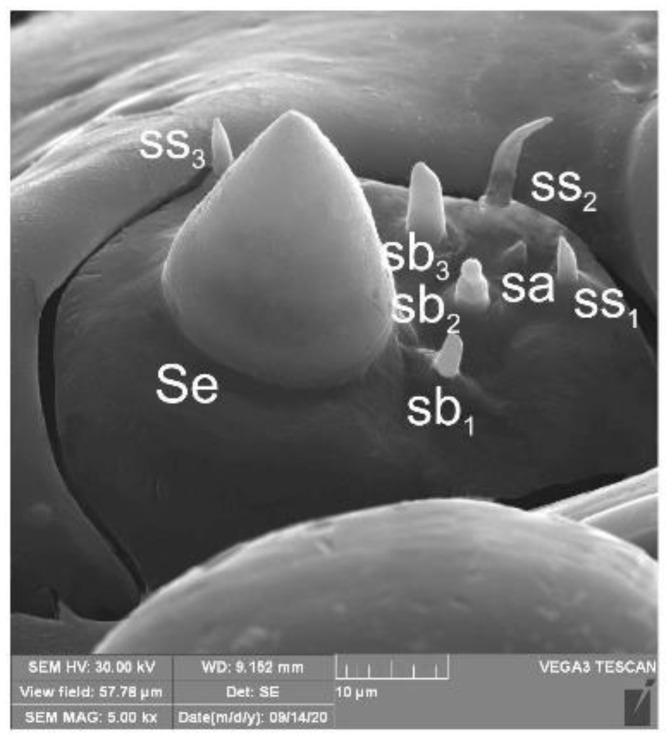
*Euryommatus mariae* mature larva, antenna (SEM micrograph) (sa—sensillum ampullaceum, Se—sensorium, ss—sensillum styloconicum, sb—sensillum basiconicum).

**Figure 8 insects-12-00151-f008:**
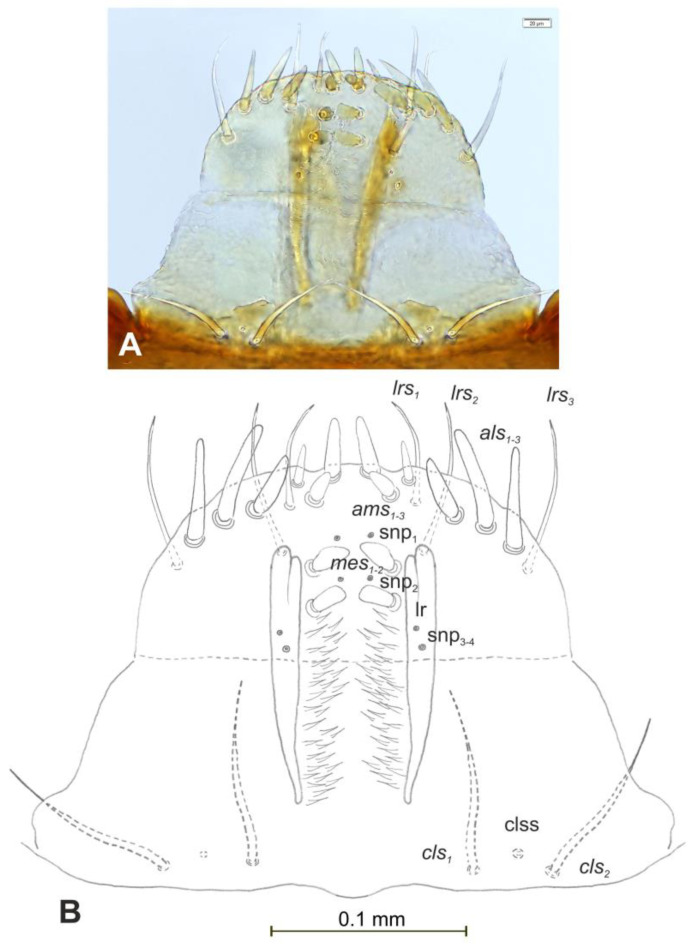
*Euryommatus mariae* mature larva, clypeus, labrum, and epipharynx. (**A**) Clypeus, labrum, and epipharynx, photo; (**B**) clypeus, labrum, and epipharynx, scheme (clss—clypeal sensorium, ds—digitiform sensillum, lr—labral rods, snp—sensillae pore, setae: *als*—anterolateral, *ams*—anteromedial, *cls*—clypeal, *lrs*—labral, *mes*—median).

**Figure 9 insects-12-00151-f009:**
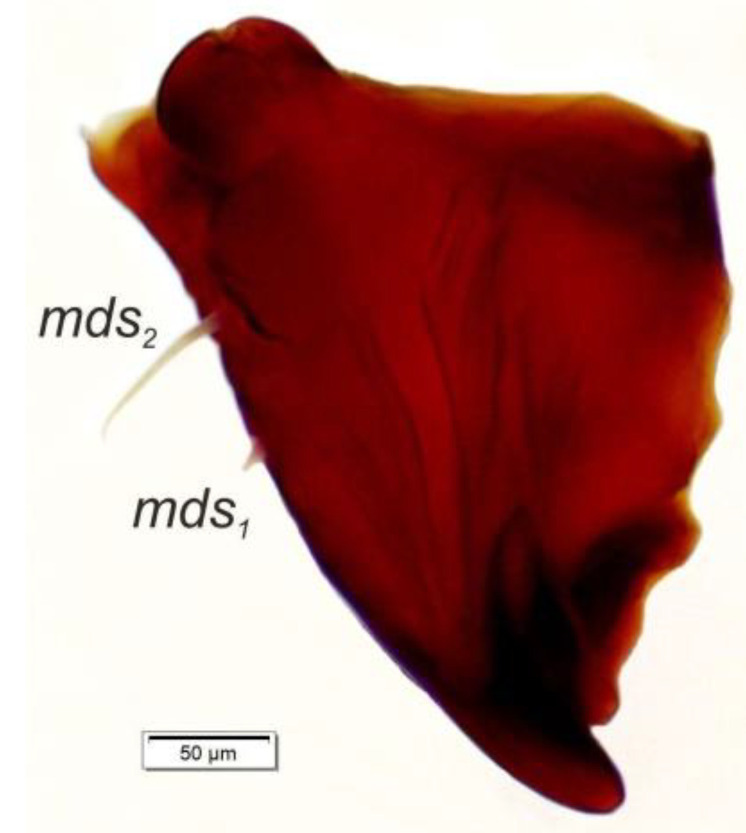
*Euryommatus mariae* mature larva, right mandible (*mds*—mandibular seta).

**Figure 10 insects-12-00151-f010:**
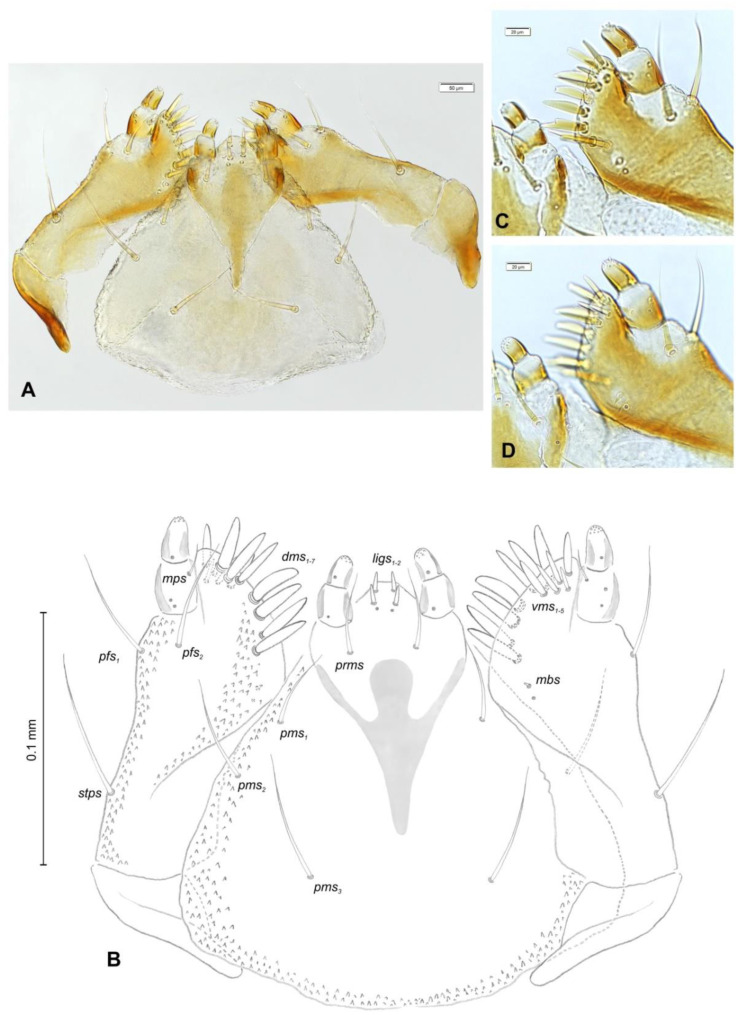
*Euryommatus mariae* mature larva, maxillolabial complex. (**A**) Maxillolabial complex, ventral view, photo; (**B**) maxillolabial complex, ventral view, scheme; (**C**) apical part of right maxilla, dorsal view; (**D**) apical part of right maxilla, ventral view, (*dms*—dorsal malar, *ligs*—ligular, *mbs*—malar basiventral, *mps*—maxillary palp, *pfs*—palpiferal, *prms*—prelabial, *pms*—postlabial, *stps*—stipal, *vms*—ventral malar).

**Figure 11 insects-12-00151-f011:**
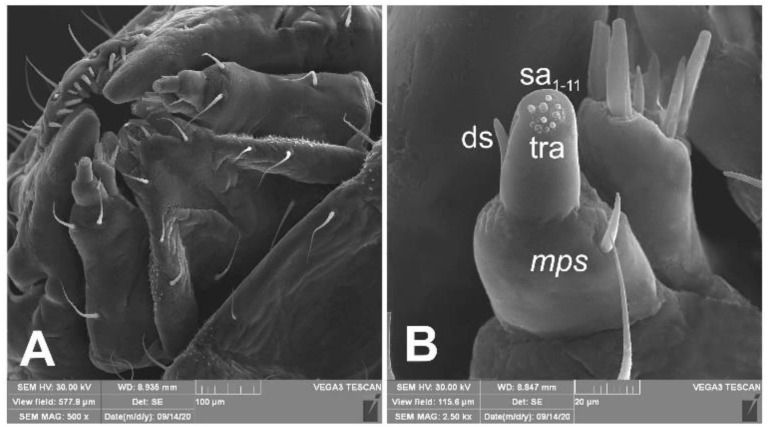
*Euryommatus mariae* mature larva, maxillolabial complex (SEM micrographs). (**A**) Maxillolabial complex, ventral aspect, SEM photo; (**B**) apical part of right maxilla, lateral aspect, SEM photo (ds—digitiform sensillum, *mps*—maxillary palp seta, sa—sensillum ampullaceum, sb—sensillum basiconicum, tra—terminal receptive area).

**Figure 12 insects-12-00151-f012:**
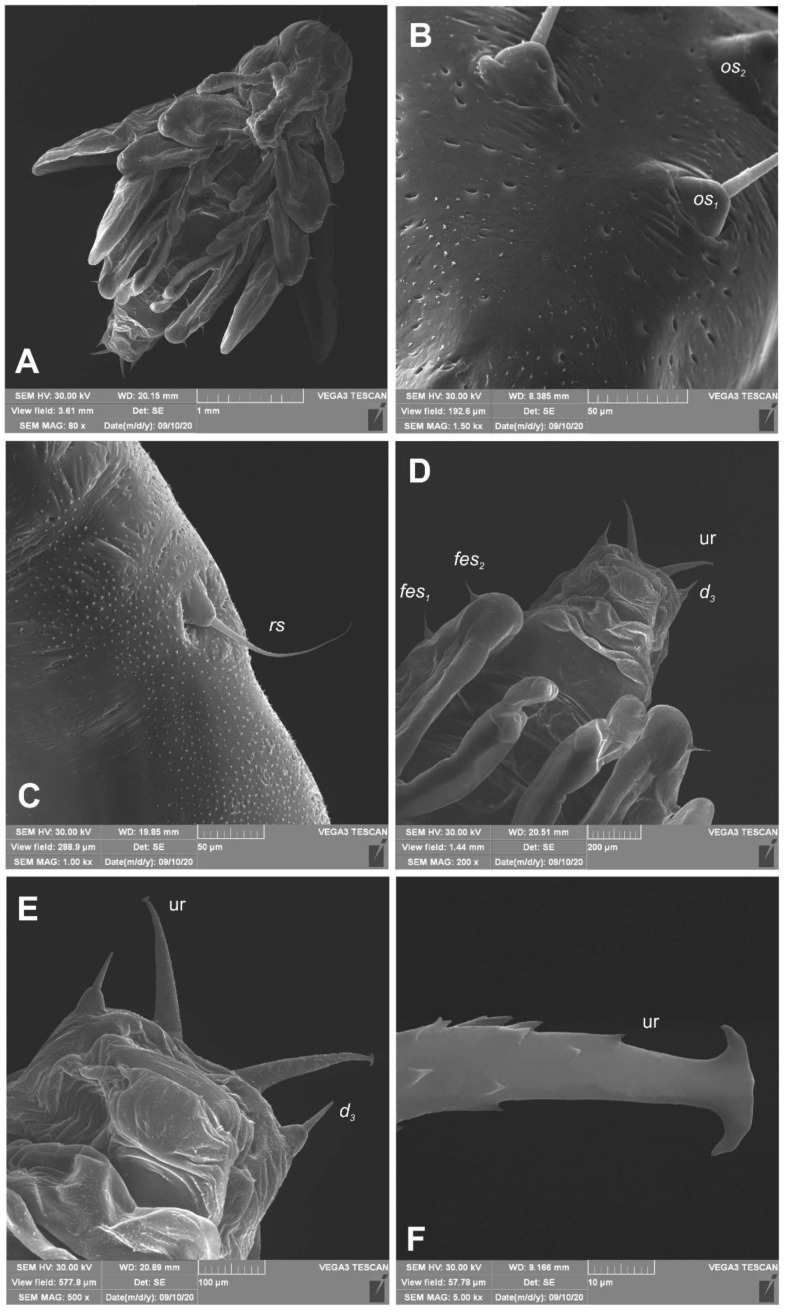
*Euryommatus mariae* pupa (SEM micrographs). (**A**) Habitus, ventral view; (**B**) head, frontal view, apical part; (**C**) rostrum base, lateral view; (**D**) abdomen, ventral view, magnification; (**E**) gonothecae; (**F**) urogomphus (ur—urogomphus, setae: *d*—dorsal, *fes*—femoral, *os*—orbital, *rs*—rostral).

**Figure 13 insects-12-00151-f013:**
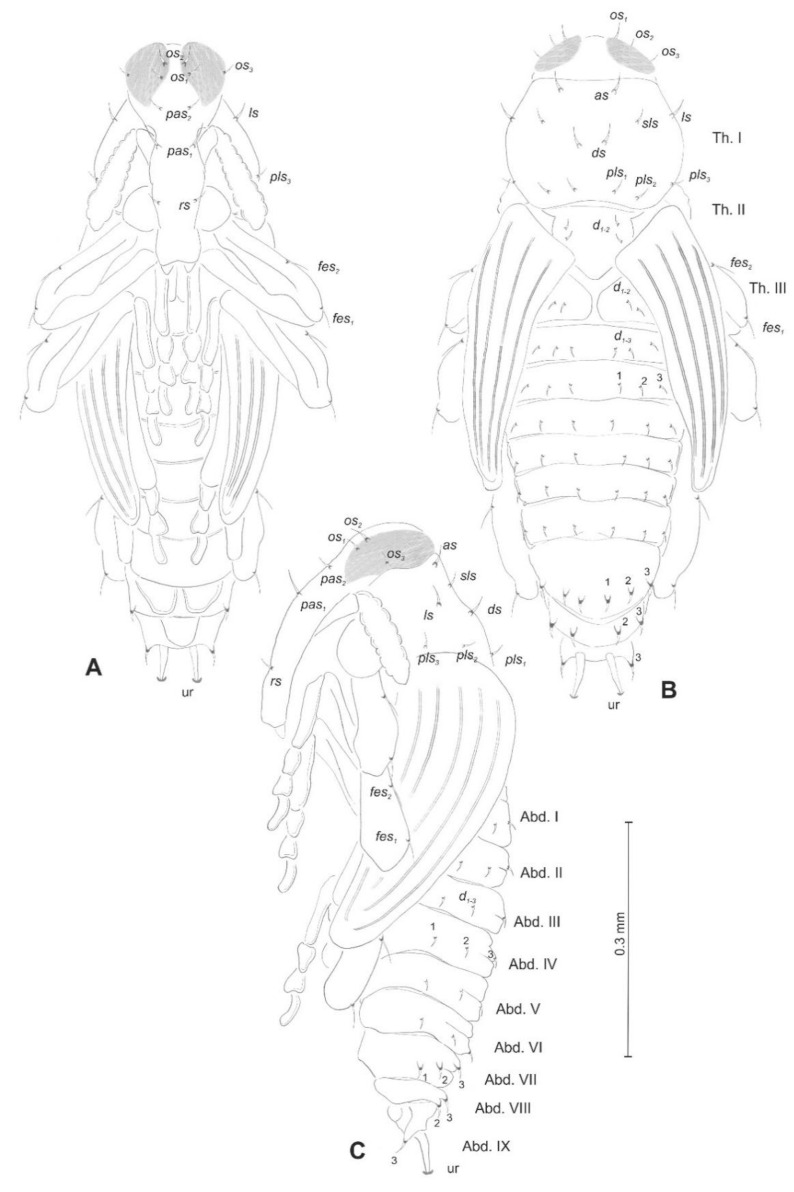
*Euryommatus mariae* pupa, habitus, and chaetotaxy. (**A**) Ventral view; (**B**) dorsal view; (**C**) lateral view (Th. I–III—pro-, meso-, and metathorax, Abd. I–IX—abdominal segments 1–9, ur—urogomphus, setae: *as*—apical, *d*—dorsal, *ds*—discal, *fes*—femoral, *ls*—lateral, *os*—orbital, *pas*—postantennal, *pls*—posterolateral, *rs*—rostral, *sls*—super lateral).

**Figure 14 insects-12-00151-f014:**
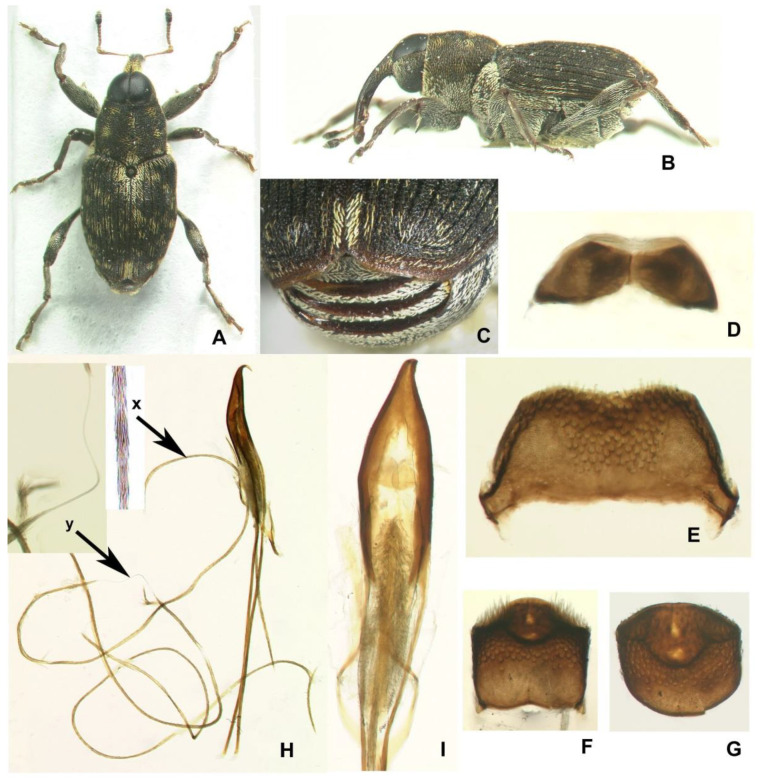
*Euryommatus mariae* adult. (**A**,**D**–**I**) Male; (**B**,**C**) female; (**A**) habitus, dorsal view (body length 3.7 mm); (**B**) habitus, lateral view (body length 4.2 mm); (**C**) margins of abdominal ventrites 2–4 posterior view; (**D**) connate hemisternites VIII; (**E**) tergite VII; (**F**) pygidium, dorsal view; (**G**) same, dorso-posterior view; (**H**) aedeagus in lateral view, showing full ejaculatory duct (broken in a few places—the specimen was processed several times) protected by a fibrous conduit (x—fibrous structure of the conduit seen under high magnification under the compound microscope; y—portion of true membranous ejaculatory duct visible after disconnection of the conduit); (**I**) aedeagus in dorsal view, showing the tegmen, orificial sclerites, and endophallic spinose brush.

**Figure 15 insects-12-00151-f015:**
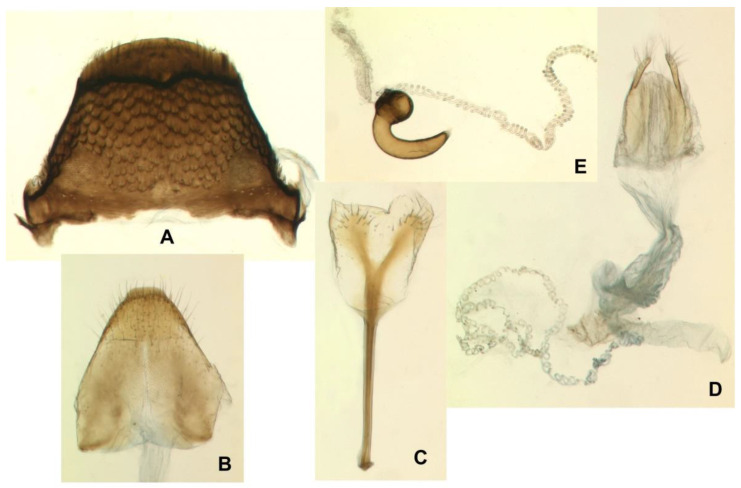
*Euryommatus mariae* adult female. (**A**) Tergite VII; (**B**) tergite VIII; (**C**) spiculum ventrale; (**D**) ovipositor, vagina, bursa copulatrix, and major portion of spermathecal duct; (**E**) spermatheca with its gland and adjoining portion of the broken spermathecal duct.

**Figure 16 insects-12-00151-f016:**
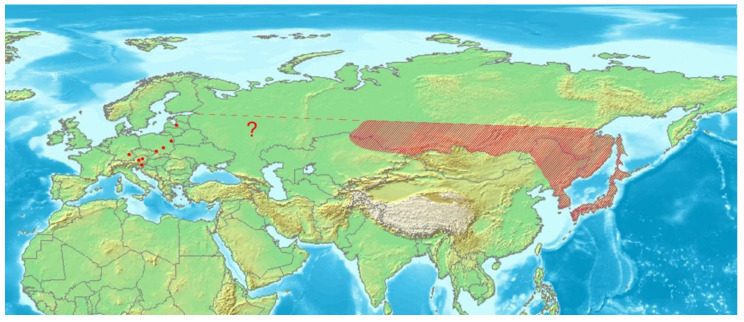
Distribution of *Euryommatus mariae* in Asia (shaded area) and Europe (dots). Map based on an image obtained from the Demis World Map Server open source (http://webmap.iwmi.org/DataSrc.htm, currently unavailable).

## Data Availability

The data presented in this study are available in this article.

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
