# Peer review of "Adult Postabdomen, Immature Stages and Biology of Euryommatus mariae Roger, 1856 (Coleoptera: Curculionidae: Conoderinae), a Legendary Weevil in Europe"

_insects, 2021, doi:10.3390/insects12020151_

Round 1
Reviewer 1 Report
This is first review, I did not indicate any comment or note anything during the text, although I read it twice.
The manuscript is written very thoroughly and it is perfect combination of specialists in different parts of examination, specialist on immature stages and one from the most respected expert on terminalia of weevils.
The paper brings three different parts, all of them very interesting: description of unknown bionomy of extremely seldom weevil, description of its immature stages and also description of very unusual terminalia of the both sexes, not focused on usual parts as the body of penis and spermatheca, but describing complete apparatus. Each from these topics can be an interesting contribution to our knowledge, combination of all three forms this paper to be exceptional.
The only part I read very insufficiently is description of immature stages, because I am not expert of this. Probably Dr. Jiří Skuhrovec could be a good authority for this part.
I can recommend this paper to publishing without any adjustment.
Reviewer 2 Report
This is an interesting manuscript that can be published after minor corrections.
- A map with the places of finds in Europe is also desirable.
- Keywords should not repeat the title of the article.
- This species lives in Siberia on Abies sibirica (see the article, Opanassenko, Kononenko, 1976).
- I collected it on Pinus silvestris.
- Some corrections see in file.

Reviewer 3 Report
This a very carefully prepared study summarizing the morphology of immature stages, new observations about biology, and some interesting new findings of adult morphology for an interesting European beetle species. The figures are especially extremely nice and illustrative - what I really like and value is the combination of careful line drawings denoting the sensilla and other structures, combined with photos of the same parts (allowing to see what future workers will really see under the microscope) and in many cases also by SEM micrographs which are of very high quality. I am not a weevil guy, so I read the paper as in "uneducated" entomologist, and have only a few comments and suggestions:
- You in fact provide a very detailed account summarizing all known data about this species. It is extremely interesting that it is a relict in Europe while widespread in the eastern Palaearctic. What about to provide the map of its distribution showing the expected extent of its continual distribution in Asia and mapping the areas from which it is known in Europe. I think such a map would nicely complete the exhaustive information you present about all other aspects of the beetle´s biology.
- I miss the general information about the habitats in which the species was collected for this study and by other authors. You only mention "spruce forest" which may refer to many different types of vegetation including the artificial timber-producing plantations so common across Europe. However, from your habitat photos, I would guess that the locality is actually a montane spruce forest, i.e. the native primary habitat, moreover of high conservation concern in central Europe. I guess this should be mentioned (along with GPS data and altitude for Cisow in the section about the locality). In this context, is the habitat preference for native montane spruce forests corresponding to the older records of this species? Are they also from mountains or areas with similar relictual native spruce forests?
- In this context, your notes on biology indicate that the beetle actually requires dead or dying trunks to be present in the forest. Maybe it would be nice to highlight in your Discussion or Conclusions what does it mean for the forest management of such localities.
- I am a little bit missing the summary of how larvae and pupae were associated with adults (and hence identified). Nothing was sequenced, so you only associated the larvae and pupae found in the twigs from the same locality with the adults, right? And in one case, you had the pupa hatching into the adult from the same locality and microhabitat. I am not saying that your identification is wrong, but it needs to be totally clear how it was done. For example, how do you guess that all larvae found are conspecific? I guess it was based on the shape of the feeding trace which was the same, and based on morphology which agreed between all these larvae, right? This should be mentioned.
- The English is kind of confusing especially in the Introduction and in the Brief Summary and should be polished.
- I am attaching the PDF with few comments directly in the text.

Reviewer 4 Report
Dear Authors, the manuscript provides excellent scientific information. I believe it will improve if a Senior Systematist will go through the manuscript and work on sentence structures as a whole. Some sentences seems informal.
26 Redundant
45 Sentence structure
49 Why is this hyphen here?
58 I am not understanding why this species is mysterious… I think “rare” might be appropriate
69 Please spell the genus when is used at the beginning of a sentence
75 Beaten, please explain this sentence, and/or provide an explanation of the collecting method
78 Please explain: (DC.) Domin.
81 Space needed?
83 SENTENCE STRUCTURE
106 What do you mind by FINDS: word choice107 Sentence structure (quite colloquial)
112 Word choice
115 Sentence structure / word choice
122 GPS needed
128 Detail needed in how measurements were taken
149 Are these needed for what reason?
170 Genitalia
171 What is hot? Nand aq?
182 Please placed mm after each measure
206 I suggest to process the image, no need to place the info at the bottom in the photo
292 Word choice
426 Thinner: provide mm, am not sure how thinner is
475 Sentence structure
484 Who has what? Please explain
497 Where is the endocarina?
510 No need -to each other-
553 ... asperities? word choice
